# Electrospun Smart Oxygen Indicating Tag for Modified Atmosphere Packaging Applications: Fabrication, Characterization and Storage Stability

**DOI:** 10.3390/polym14102108

**Published:** 2022-05-21

**Authors:** Shivam Panwar, Narender Raju Panjagari, Ashish Kumar Singh, Gaurav Kr Deshwal, Richa Badola, Prashant Saurabh Minz, Gulden Goksen, Alexandru Rusu, Monica Trif

**Affiliations:** 1Dairy Technology Division, ICAR-National Dairy Research Institute, Karnal 132001, Haryana, India; shivampanwarndri@gmail.com (S.P.); aksndri@gmail.com (A.K.S.); ndri.gkd@gmail.com (G.K.D.); richa.ucals@gmail.com (R.B.); 2Dairy Engineering Division, ICAR-National Dairy Research Institute, Karnal 132001, Haryana, India; psminz@gmail.com; 3Department of Food Technology, Vocational School of Technical Sciences at Mersin Tarsus Organized Industrial Zone, Tarsus University, Mersin 33100, Turkey; guldengoksen@tarsus.edu.tr; 4Life Science Institute, University of Agricultural Sciences and Veterinary Medicine Cluj-Napoca, 400372 Cluj-Napoca, Romania; 5Animal Science and Biotechnology Faculty, University of Agricultural Sciences and Veterinary Medicine Cluj-Napoca, 400372 Cluj-Napoca, Romania; 6Food Research Department, Centre for Innovative Process Engineering (CENTIV) GmbH, 28816 Stuhr, Germany; monica_trif@hotmail.com

**Keywords:** carrageenan, oxygen indicator, electrospun, Taguchi, intelligent packaging, stability

## Abstract

Pack integrity is essential for the success of modified atmosphere packaging of food products. Colorimetric oxygen leak indicators or tags are simple and smart tools that can depict the presence or absence of oxygen within a package. However, not many bio-based electrospun materials were explored for this purpose. Ultraviolet light-activated *kappa*-carrageenan-based smart oxygen indicating tag was developed using the electrospinning technique in this study and its stability during storage was determined. *Kappa*-carrageenan was used with redox dye, sacrificial electron donor, photocatalyst, and solvent for preparing oxygen indicating electrospun tag. Parameters of electrospinning namely flow rate of the polymer solution, the distance between spinneret and collector, and voltage applied were optimized using Taguchi L9 orthogonal design. Rheological and microstructural studies revealed that the electrospinning solution was pseudoplastic and the mat fibers were compact and non-woven with an average fiber size of 1–2 microns. Oxygen sensitivity at different oxygen concentrations revealed that the tag was sensitive enough to detect as low as 0.4% oxygen. The developed tag was stable for at least 60 days when stored in dark at 25 °C and 65% RH. The developed mat could be highly useful in modified atmosphere packaging applications to check seal integrity in oxygen devoid packages.

## 1. Introduction

Demand for modified atmosphere packaging (MAP) and intelligent packaging (IP) systems is increasing at a rapid rate. Market intelligence companies have predicted that by the year 2025, the smart food packaging market is going to dominate North America followed by Europe and the Asia Pacific region, while by the year 2026, the MAP market is projected to reach US $ 18.51 billion with a CAGR of about 6.2% [1,2]. In MAP, the gas composition in the headspace of a food package is changed in such a way that it leads to extended shelf life by reducing microbial growth and chemical deterioration reaction [3]. IP contains external or internal indicators that portray historical information of the package along with or without the quality of the food. IP does not help in enhancing the shelf life of the packaged food products but serves as a mirror to the inside package conditions and informs the consumer about the same. IP systems comprise time-temperature indicators, critical temperature indicators, and leak indicators [4,5].

One of the major requirements of MAP is to maintain seal or package integrity. The atmosphere inside the package can be changed by either tampering, damage during storage and transportation, or permeation of air through packaging material [6]. For any of these reasons, oxygen concentration increases inside the package and spoils the food through enzymatic and nonenzymatic reactions, lipid oxidation, oxidation of flavors, and destruction of ascorbic acid. Further, aerobic spoilage organisms’ growth also increases. As and when headspace oxygen increases due to atmospheric air ingress, food’s deteriorative reaction rate increases. Hence, to check the efficacy of MAP, gas leak indicators or sensors can be used [7]. Most animal-origin foods including meat, fish, and dairy products are MAP packaged in oxygen-deprived atmospheres. Hence, the integration of oxygen sensors or leak indicators with such MAP food products can serve as a quality and safety indicator.

Different types of sensors or indicators have been used for detecting the headspace oxygen content of packaged foods. However, their operating cost is very high, they lack portability, and they require skilled manpower for operation. Hence, simple and low-cost visual IP systems or indicators are required. A visual colorimetric oxygen indicator works by sensing the headspace oxygen concentration and indicating the status through a visual color change that could be perceived with a naked eye [8,9]. Such indicators can be manufactured in the form of a printed label, a tablet, a roll-to-roll patterned indicator, or a polymeric film that could be used with the incorporation of suitable additives [10,11]. Based on various techniques and principles, different materials have been used as sensing materials to detect the presence of targeted gases. Different methods are available for fabricating gas leak indicators such as casting, inkjet-printed, extrusion, doctor blade, and electrospinning [11,12].

A gas sensor is a device that detects the presence of different gases in an area, especially gases that might be critical or harmful to industrial operations or humans or animals. During the last decade, nanoparticles have attracted considerable scientific interest due to their size and shape-dependent properties that can be tailored for a wide range of applications. Polymer-based nanostructured materials have also been used as smart biopolymers for sensing gases, pathogens, and biomolecules as well as other sensor applications [13,14,15]. Naturally occurring materials such as metal oxides, carbon nanotubes, and graphene in their nanosizes or materials designed to be of an ultrathin size such as electrospun structures made from polyaniline, titanium oxide, etc. have been attempted as gas sensors [12,16,17,18,19].

Electrospinning is an emerging versatile technique used for fabricating micro and nano-sized fibers continuously from polymer solutions through a micro-syringe pump in the presence of very high voltage. It provides a three-dimensional structured membrane having ultrafine fibers with a higher surface area [12,20,21,22]. To electrospin nonpolymeric molecules, different conductive polymers are mixed to have high viscosity and intermolecular entanglement in the solution. Large surface area, controllable thickness, and ultrafine structures are characteristics of electrospun fibers and are major factors for such structures as sensing materials. Several electrospun structures with improved detection limits have been used to fabricate extremely sensitive gas sensors [12,23]. Their high sensitivity and reduced response time compared with continuous thin films could be due to enhanced surface area, which makes them highly suitable for sensing applications. In the recent past, gas sensors have been developed using electrospun nanofibers to detect NH_3_, H_2_S, H_2_, alcohol vapor, biogenic amines, and others [23,24,25,26,27]. Recently, Mohammadi and coworkers [28] reviewed the application of the electrospinning technique in the development of pH-based intelligent food packaging systems. Earlier, electrospun oxygen indicators were developed using polyethylene oxide [29], polyvinyl alcohol [30], and sulfonated electrospun polystyrene [31] fibers. However, polysaccharide biopolymers from renewable sources such as microbial (e.g., xanthan, pullulan; bacterial cellulose; pectin) and seaweed-based (e.g., furcellaran, alginate, carrageenan) gums, which could usher in sustainable solutions [32,33,34,35,36] to the food processing and packaging industry, have not been particularly explored for developing electrospun oxygen indicators.

Based on the above, it is clear that despite the need for a simple, highly sensitive, colorimetric, and low-cost oxygen leak indicator, only limited studies have focused on adopting the electrospinning technique for fabricating biopolymer-based oxygen leak indicators [30,31] for food packaging applications. Hence, an attempt was made to fabricate and characterize a carrageenan-based electrospun ultraviolet light-activated colorimetric oxygen indicator tag for MAP foods.

## 2. Materials and Methods

### 2.1. Materials

Carrageenan (predominantly kappa and lesser amounts of lambda; faint beige colored powder; particle size: 200 mesh; melting point: 50–70 °C; pH (10% *w/v* solution): 9; moisture: 10%; fat: 0.30–0.50%; CAS number: 9000-07-1; product number: C1013-500G; lot number: SLBK3896V), polyethylene oxide (PEO) (average molecular weight: 100,000 daltons; CAS number: 25322-68-3; lot number: MKBZ1708V), and resazurin sodium salt (dark grey colored powder; dye content: minimum 75%; visual color at pH 3.8 and 6.5: orange and violet, respectively; molecular weight: 251.17; CAS number: 62758-13-8) were obtained from Sigma-Aldrich, St. Louis, MO, USA. Titanium dioxide (purity: minimum 99%) was obtained from Sisco Research Laboratories, Mumbai (India). Glycerol (purity: minimum 99.5%; molecular weight: 92.09 g/mol) was procured from Merck Specialities Private Limited, Mumbai (India). Ethanol (purity: minimum 99.9%) was obtained from Changshu Hongsheng Fine Chemical Co. Ltd., Changshu City, Jiangsu Province (China).

### 2.2. Characterization of Electrospinning Solution and Optimization of the Electrospinning Process

Knowledge about the behaviors of electrospinning solutions containing different components is essential for process optimization at a later stage. Hence, viscosity and flow behavior (shear rate 0 to 100 s^−1^) of a kappa-carrageenan-based electrospinning solution containing glycerol, titanium dioxide, and PEO were determined at 25 °C, 35 °C, and 45 °C using a rheometer (MCR52, Anton-Paar, Germany) supported by software (Rheoplus/32, V.3.61, Anton-Paar, Graz, Austria). Three commonly used rheological models for describing biopolymer-based solutions were fitted to the data from Equations (1)–(3). The electrical conductivity of each solution was determined at 20, 30, and 40 °C using a handheld electrical conductivity meter (ERMA, INC), and observations were recorded in Siemen.cm^−1^. Each time, a fresh sample was used, and mean values of triplicate data were presented.

(1)
$$\mathrm{Power}-\mathrm{law}\;\mathrm{model}:\;Ԏ=K\cdot \gamma^{n}$$


(2)
$$\mathrm{Herschel}\;\mathrm{Bulkley}\;\mathrm{model}:\;Ԏ={Ԏ}_{o}+K\cdot \gamma^{n}$$


(3)
$$\mathrm{Casson}\;\mathrm{model}:\;\sqrt{Ԏ}=Koc+Kc\sqrt{\gamma }$$


For the production of oxygen-indicating tag, one gram of carrageenan was dissolved in 90% ethanol (*w*/*v*) and glycerol (7.2%, *w*/*v*), and titanium dioxide (7.2%, *w*/*v*) was added along with PEO (20%, *w*/*v*). The quantity of carrageenan was chosen from Deshwal and others [7]. The selected dye (resazurin sodium salt) was dissolved in a small amount of solvent that was later mixed with the biopolymer solution. The prepared solutions were subjected to continuous stirring at ambient temperature for 15 min using a magnetic stirrer at about 600 rpm. The prepared solution was later filled into a 2 mL syringe followed by loading into an electrospinning machine (Model: Espin Nano V2, supplied by M/s Physics Equipments Company, Chennai, India). The negative electrode was attached to the syringe needle, while the positive electrode was attached to the collector drum over which aluminum foil was wrapped for collecting the fiber mat. Desirable temperature (45 °C) and relative humidity (30%) of the electrospinning chamber were maintained followed by adjusting the process parameters. The collector was adjusted to 1000 rpm before starting the electrospinning process. Optimization of electrospinning process parameters (independent factors), namely, the applied voltage (8–12 kV), the flow rate of the carrageenan solution (0.2–0.6 mL/h), and the distance between the collector and syringe needle (10–15 cm), was done using Taguchi’s orthogonal design L9 (Figure 1). At the end of the process, the developed electrospun mat was peeled from the aluminum foil and kept in a polypropylene tray at room temperature for storage. Before use as a tag, the peeled mat was cut into the desired size of approx. 1 in^2^. It is well-known that the electrospinning process parameters affect the resultant fiber diameter. In the present case, it may affect the ability of the fibers to sense the presence of oxygen. The aim is to obtain fibers with two-pronged benefits: that they will change their color upon ultraviolet light treatment in absence of oxygen (air) and that the activated fibers will respond to even a small quantity of oxygen with even a slight but distinguishable color change. Because of this, two responses were considered dependent factors for optimization. One of them is the total color difference (*ΔE*) between the ultraviolet light-activated tag that regained its original color post-exposure to oxygen or ambient air (recovered) and the un-activated (original) tag (*∆E_(r-o)_*); the other is between the ultraviolet light (photo) activated and the original tag (*∆E_(p-o)_*). Details of the ultraviolet light activation of the electrospun mat/tag and the formulae for calculating *∆E_(r-o)_* and *∆E_(p-o)_* are given in the next section. For determining the optimum process parameters that minimize *∆E_(r-o)_* and maximize *∆E_(p-o)_*, Taguchi’s “smaller the better” and “bigger the better” criteria were used, respectively, in Equations (4) and (5):
(4)
$$\frac{\;S}{N}=-10\log(\frac{1}{n}\sum_{i=1}^{n}y_{i}^{2})$$


(5)
$$\frac{\;S}{N}=-10\log(\frac{1}{n}\sum_{i=1}^{n}\frac{1}{y_{i}^{2}})$$

where *S*/*N* is the signal-to-noise ratio, *n* is the number of observations, and *y* is the respective *ΔE*.

### 2.3. Characterization of Electrospun Oxygen Indicating Tag

#### 2.3.1. Determination of the Color Values of Digital Images of Electrospun Tags

Electrospun tags were placed in nylon pouches followed by vacuum packaging to remove the oxygen in the pouches. Later, they were subjected to ultraviolet (UV) treatment for photo-activation to make them oxygen sensitive. The electrospun tag-containing nylon pouches were kept at a distance of 10–15 cm from the UV tubes (M/s Philips) for activation. The UV-activated electrospun tags were then ready for oxygen detection. The color of the electrospun tag was determined using a digital camera. The original (non-UV activated), photo-activated (UV-treated), and recovered (UV-treated but regained upon exposure to oxygen or ambient air) electrospun tag’s color was determined in terms of *L, a,* and *b* using the Adobe Photoshop CS6 software. These values were converted into *L*, a*,* and *b** (lightness, redness-to-greenness, yellowness-to-blueness), as they are device-independent, and cover a larger range than RGB and CMYK systems [37], using the formulas in Equations (6)–(8) given by Kumari and coworkers [38]. The time required for visual color change in the tag (photo-activation) was also recorded. The total color difference between the recovered and original electrospun tag (*∆E_(r-o)_*) and between the photo-activated and original electrospun tag (*∆E_(p-o)_*) were determined using Equations (9) and (10).

(6)
$$\;L^{*}=\frac{L\times 100}{255}$$


(7)
$$a^{*}=\frac{240\times a-120\;}{255}$$


(8)
$$b^{*}=\frac{240\times b-120\;}{255}$$


(9)
$${∆E}_{\left(p-o\right)}=\sqrt[{2}]{{\left(L_{r}^{*}-L_{o}^{*}\right)}^{2}+{\left(a_{r}^{*}-a_{o}^{*}\right)}^{2}+{\left(b_{r}^{*}-b_{o}^{*}\right)}^{2}}$$


(10)
$${∆E}_{\left(p-o\right)}=\sqrt[{2}]{{\left(L_{p}^{*}-L_{o}^{*}\right)}^{2}+{\left(a_{p}^{*}-a_{o}^{*}\right)}^{2}+{\left(b_{p}^{*}-b_{o}^{*}\right)}^{2}}$$

where *L*, *a,* and *b* are the color values of images obtained from the software while *o, p, and r* subscripts refer to original, photo-activated, and recovered, respectively.

#### 2.3.2. Moisture Sorption Behavior of the Electrospun Tag

The moisture absorption behavior of the electrospun mat indicates the storage stability of the spun mats. Hence, the gravimetric (sorbostat) method was adopted for determining the moisture sorption behavior of the fabricated tag in the water activity range of 0.11 to 0.92 at 25 °C using saturated salt solutions [39]. Sorption data were fitted to the BET Equation (11), GAB Equation (12), Park Equation (13), Ferro-Fontan Equation (14), Peleg Equation (15), and D’Arcy & Watt Equation (16) [40,41,42,43,44,45] models given below. Models that met multiple criteria, namely, the coefficient of determination (*R*^2^), root mean square percent (*RMS%*), and percent mean deviation (*P%*) were considered to have the best fit.

(11)
$$\mathrm{BET}\;\mathrm{model}\;\frac{M}{M_{o}}=\frac{C_{b}a_{w}}{\left(1-a_{w}\right)}\left(1-a_{w}+C_{b}a_{w}\right)$$


(12)
$$\mathrm{GAB}\;\mathrm{model}\;M=\frac{M_{o}CKa_{w}}{\left(1-Ka_{w}\right)}\left(1-Ka_{w}+CKa_{w}\right)$$


(13)
$$\mathrm{Park}\;\mathrm{model}\;C=\frac{A_{L}b_{L}a_{w}}{1+b_{L}a_{w}}+K_{H}a_{w}+K_{a}na_{w}^{n}$$


(14)
$$\mathrm{Ferro-Fontan}\;\mathrm{model}\;M={\left[\frac{B}{Ln\left(A/a_{w}\right)}\right]}^{C}$$


(15)
$$\mathrm{Peleg}\;\mathrm{model}\;M=A{\left(a_{w}\right)}^{C_{1}}+B{\left(a_{w}\right)}^{C_{2}}$$


(16)
$$D’\mathrm{Arcy}\;\&\;\mathrm{Watt}\;\mathrm{model}\;M=\frac{K_{1}K_{2}a_{w}}{1+K_{1}a_{w}}+K_{3}a_{w}+\frac{K_{4}K_{5}a_{w}}{1-K_{4}a_{w}}$$


#### 2.3.3. Global Migration into Food Simulants

Since the intended application of the oxygen-indicating mat is packaged food products, its overall or global migration, in case of direct contact, into food products assumes importance. Hence, global migration from the fabricated electrospun tag was determined by adopting the method of the Bureau of Indian Standards [46], which was adopted from the European Union standard, in triplicate. The kappa-carrageenan-based tag was exposed to different food simulants on both sides with a total surface area (*A*) of 50 cm^2^ (5 cm × 5 cm × 2 sides). At the end of the stipulated duration, simulants were evaporated, and the mass of residue (*M*) was calculated in mg by subtracting the blank value. The extractive was calculated per Equation (17) and expressed in mg/dm^2^. Heptane extractive values were divided by five (factor) for arriving at the extractive.

(17)
$$\mathrm{Amount}\;\mathrm{of}\;\mathrm{extractive}=\frac{M}{A}\;\times \;100\;\mathrm{mg}/{\mathrm{dm}}^{2}\;$$


#### 2.3.4. Microstructure and Oxygen Sensitivity

The electrospun tag was subjected to scanning electron microscopy (M/s Hitachi-High-Tech India Pvt. Ltd., Gurugram, India) to determine the microstructure. The sensitivity of the kappa-carrageenan-based electrospun oxygen indicator tag was tested by opening the seal of the vacuum pack containing the oxygen indicator and exposing it to atmospheric oxygen. Similarly, the sensitivity of the oxygen indicator at low oxygen concentration (0.3%) was determined by MAP of an empty polystyrene tray sealed with nylon film attached with an oxygen indicator tag. The modified atmosphere packaged tray was made photo-active using the UV treatment system as mentioned in an earlier section, and the time taken by the photo-activated indicator to recover to its original color was recorded.

### 2.4. Storage Stability of Electrospun Tag

To determine the storage stability, electrospun oxygen tags were exposed to light and dark conditions, at 5 °C and 25 °C and at 65% and 95% relative humidity (RH). For light stability, tags were placed in casting trays in triplicate. One of the trays was wrapped with aluminum foil to provide dark conditions, and the other tray was exposed to 12 h of light and 12 h of darkness without aluminum foil wrapping (at 25 °C). For temperature stability, trays containing tags were placed in cabinets maintained at 5 °C and 25 °C. Finally, for humidity stability, trays containing tags were placed in desiccators maintained at 65% and 95% RH using appropriate saturated salt solutions at 25 °C. Except for the images of stored tags exposed to 95% RH, images of other tags were captured using a digital camera for 60 days at an interval of 2 days, and then the color values were estimated. For the tags exposed to 95% RH, images were captured at an interval of 30 min. The total color difference (*∆E*) in tags was calculated using below equation given below:
(18)
$$∆E=\sqrt{{\left(L_{o}^{*}-L_{t}^{*}\right)}^{2}+{\left(a_{o}^{*}-a_{t}^{*}\right)}^{2}+{\left(b_{o}^{*}-b_{t}^{*}\right)}^{2}}$$

where subscript *o* refers to the color reading of the tag on the 0th day used as the reference and *t* refers to the reading on a particular day or interval.

### 2.5. Statistical Analysis

The data obtained in the study were subjected to a one-way analysis of variance (ANOVA) at a 5% level of significance to draw meaningful conclusions using JMP software (ver. 10) (JMP Singapore, Singapore).

## 3. Results and Discussion

### 3.1. Viscosity and Flow Behavior Properties of Electrospinning Solution

With an increase in the shear rate from 10 to 100 s^−1^, apparent viscosity was found to decrease at all three temperatures (Figure 2A). At a very low shear rate (10 s^−1^), the viscosity of the sample was found to decrease from 2.52 Pa·s to 2.18 Pa·s when the temperature was increased to 45 °C (*p* < 0.05). At a higher shear rate (100 s^−1^), viscosities dropped to 1.89–1.92 Pa·s. However, no significant change in the viscosities was observed with changes in the temperature (*p* > 0.05). It can be seen from Figure 2A that all the solutions exhibited shear thinning behavior. Commercial carrageenan solutions usually have viscosities in the range of 5–800 mPa·s (0.005–0.8 Pa·s) at 1.5% concentration (*w*/*w*) and 75 °C, and they generally exhibit pseudoplastic (shear thinning) properties [47]. In the present study, the higher apparent viscosities could be due to the determination of viscosity at lower concentrations of kappa-carrageenan (1%), the lower temperature (25 °C–45 °C), and the presence of glycerol and PEO.

The data obtained from the shear stress-shear strain experiments were fitted to power-law, Herschel Bulkley, and Casson models to determine the flow behavior (*n*) and consistency (*K*) indices (Figure 2B). The power-law model was revealed to have the best fit, with a coefficient of determination (*R*^2^) ranging between 0.9986 and 0.9997 (Figure 2B). According to the power-law model, the flow behavior indices (*n*) were 0.923, 0.965, and 0.960, and the consistency coefficients were 2.701, 2.223, and 2.238 at 25 °C, 35 °C, and 45 °C, respectively. Elfak and others [48] reported that the relative viscosity of kappa-carrageenan solution increased with high concentrations of added solutes (glucose and sucrose) and that the solution displays non-Newtonian behavior, which could be due to kappa-carrageenan-specific solute-solute interactions. Nishinari and Watase [49] observed that the hydrogen bonding between hydroxyl groups in kappa-carrageenan and glycerol stabilize the solution. In the present study, the coefficient of determination (*R*^2^) of the power-law model was found to be high at all three temperatures, confirming the non-Newtonian behavior. The flow behavior indices (*n*) of the kappa-carrageenan-based solution were found to be 0.923, 0.965, and 0.960 at 25, 35, and 45 °C, respectively. For the Hershel-Bulkley model, *n* was found to be 1.392 (25 °C), 1.449 (35 °C), and 1.536 (45 °C) with *R*^2^ ranging between 0.9701 and 0.9770. For the Casson model, *n* was found to be 1.088 (25 °C), 1.228 (35 °C), and 1.222 (45 °C) with *R*^2^ ranging between 0.9145 and 0.9578.

### 3.2. Electrical Conductivity of Electrospinning Solution

A significant increase (*p* < 0.05) in the electrical conductivity of the kappa-carrageenan-based electrospinning solution was observed with an increase in the temperature. The electrical conductivity was found to be 54.17 µs·cm^−1^, 94.9 µs·cm^−1^, and 141.5 µs·cm^−1^ at 20 °C, 30 °C, and 40 °C, respectively. Lu and coworkers [50] reported that the electrical conductivities of sodium alginate and PEO-containing electrospinning solution were, respectively, 1.01 ms·cm^−1^ and 1.24 ms·cm^−1^ for 1:1 and 1:3 ratios of sodium alginate and PEO-containing solutions. Alborzi and coworkers [51] studied the electrical conductivity of a sodium alginate, pectin, and PEO-based solution and reported it as 5.52 ms·cm^−1^. Further, it was reported that with an increase in the PEO concentration, there was a decrease in the electrical conductivity. Horuz and Belibagli [52] reported that charge density on the surface of the drop on the tip of the needle increases with an increase in electrical conductivity, which helps in the formation of a thin diameter fiber due to enhanced elongation forces being exerted on the polymer jet. Hence, in this study, to obtain a small fiber and also to maintain a lower viscosity of the solution, electrospinning trials were carried out at 45 °C.

### 3.3. Optimization of Electrospinning Process Parameters

Fabrication and optimization were carried out using Taguchi’s orthogonal array design based on the total color difference between the original (non-UV treated) tag and the photo-activated tag (UV-treated) and the total color difference between the original tag and the recovered tag (tag exposed to ambient air after UV-treatment). Based on the number of process parameters chosen and their levels, Taguchi’s orthogonal design revealed nine sets of combinations. The physical appearances of the original, photo-activated, and recovered kappa-carrageenan-based electrospun leak indicator tags of the nine different electrospinning parameter combinations are shown in Figure 1. It can be seen that the color of the original tags differed from the color of the photo-activated tags, and the color of the original tags was observed to be similar to that of recovered tags for all the combinations. The measured color values of original (*L_o_*, a_o_** and *b_o_**), photo-activated (*L_p_*, a_p_** and *b_p_**) and recovered (*L_r_*, a_r_** and *b_r_**) kappa-carrageenan-based electrospun tag in terms of *L*, a*,* and *b** is presented in Table 1. The table reveals significant differences (*p* < 0.05) in the lightness values of the original tags (*L_o_**) before photo-activation, with the highest value observed for A3B3C2 (19.78) and the lowest for A1B2C2 (12.13). UV activation led to significant differences (*p* < 0.05) in lightness (*L_p_**) values, with the highest (18.82) and lowest (11.64) values associated with A3B3C2 and A1B2C2. Similarly, the lightness values of UV-activated tags were found to differ significantly (*p* < 0.05) upon exposure to air (recovered) (*L_r_**), with the highest (20.02) and the lowest (13.38) being recorded for A3B1C3 and A1B2C2 batches. The changes in the lightness values of the original (*L_o_**) and photo-activated (*L_p_**) tags between A2B1C2 and A3B1C3 were significant (*p* < 0.05). However, *L_o_** and *L_r_** values for the A2B1C2, A2B3C1, and A3B1C3 treatments were not significantly different *(p* > 0.05).

It can be seen from Table 1 that significant changes in *a** values of the non-UV activated, photo-activated, and recovered tags were recorded (*p* < 0.05). A3B2C1 and A1B1C1 combinations recorded the lowest (1.85) and highest (5.65) values among the nine combinations of original (*a_o_**) tags. Similarly, among the photo-activated tags, A2B1C2 and A3B3C2 resulted in significantly lower (8.91) and significantly higher (11.91) redness (*a_p_**) values, whereas A3B2C1 and A2B2C3 recorded the lowest and highest values for recovered tag (*a_r_**). Deshwal and coworkers [7] reported that the *a** values of the original and the UV-activated kappa-carrageenan and methylene blue-based films were significantly different (*p* < 0.05), but the authors reported nonsignificant (*p* > 0.05) changes in *a_o_** and *a_r_** tags among all the treatments.

The blueness values of the original and pre-photo-activated tags (*b_o_**) of different combinations were found to be significantly different. The *b_o_** values of tags for the treatments A2B1C2 (−13.53) and A2B3C1 (−17.12) were the highest and lowest, respectively. However, only the changes between *b_o_** and *b_r_** of A2B3C1 and A3B1C3 were found to be significantly different (*p* < 0.05) among all nine combinations. The *b_o_** and *b_r_** values for combinations A1B1C1 (−14.26), A1B2C2 (−15.06), A2B2C3 (−14.32), A2B3C1 (−13.50) A3B2C1 (−14.29), and A3B3C2 (−15.09) were not significantly different.

The kappa-carrageenan-based electrospun tag changes color from blue to violet upon UV activation and recovers to its original blue color when exposed to an oxygen-rich environment. This color change is attributed to photo-activation because upon UV treatment, semiconductor-like titanium oxide produces photo-generated electrons that are displaced from their valence band to the conduction band, thereby reducing the redox dye color to its bleached form, which changes the color [53]. Similarly, Vu and Won [54] fabricated carrageenan and methylene blue-based colorimetric oxygen indicator film and revealed that the film was discolored when exposed to UV light and regained its color when exposed to oxygen. They also concluded that a decrease in *a** occurred when the photo-activated film was kept in an oxygen environment to recover the color. In this study, the changes in the color values with the electrospinning process parameters could be attributed to changes in the resultant fiber diameter. Horuz and Belibagli [52] reported that with changes in voltage, flow rate, and distance during the electrospinning process, the fiber morphology of the electrospun tag changes. Similarly, Mihindukulasuriya and Lim [29] reported that during the preparation of the PEO-TiO_2_ nanoparticle-based oxygen indicator, increased fiber diameter was observed due to increased ethanol concentration, which ultimately caused an increased color-absorbance-to-color-scattering ratio. Further, it was reported that the altered ratio aids in the better color recovery of oxygen indicator when it is exposed to air compared with the small fiber diameter-based oxygen indicator.

The total color difference values (*ΔE*) were calculated based upon the findings that the *L**, *a**, and *b** values between the original (non-UV activated) and recovered tags (*ΔE_(r-o)_*) among A1B3C3 (4.69), A2B3C1 (4.62), and A3B1C3 (7.47) were significantly different (*p* < 0.05). The total color difference values between the photo-activated and original tags (*ΔE_(p-o)_*) of A3B2C1 and A3B3C2 were found to be significantly (*p* < 0.05) higher among all the treatments. Human eyes can detect differences in color if the instrumental color difference values are more than 5.0 [55]. Thus, higher values of total color difference between original and photo-activated tags (*ΔE_(p-o)_*) are desirable for completely distinguishing between oxygen-sensitive and non-activated (original) tags, whereas smaller total color differences between original and recovered tags (*ΔE_(r-o)_*) are desirable for complete recovery of color. Table 1 reveals that *ΔE_(p-o)_* of the electrospun tags was found to be more than 5 for all the treatments, clearly distinguishing between non-activated and activated tags. Similar results of *ΔE* values increasing when exposed to UV light for a nano titanium oxide-based UV-activated oxygen bio-indicator have been reported by Khankaew and coworkers [56]. The total color difference between the recovered and original tags (*ΔE_(r-o)_*) was found to be less than 5 for all treatments except A3B1C3, indicating recovery of tag color upon exposure to an oxygen environment. The lowest total color difference between recovered and original tags was 2.53 for treatment A1B2C2.

Electrospinning process parameters were optimized considering the aberration of signal-to-noise (S/N) ratio values from the desired values, which were determined with the aid of software according to the smaller the better characteristic formula for total color difference between original and recovered tags (*ΔE_(r-o)_*) and the bigger the better characteristic formula for the total color difference between the original and photo-activated tags (*ΔE_(p-o)_*). Figure 3 reveals that 8 kV voltage, 0.4 mL/h flow rate, and 10 cm distance with a desirability value of 0.97 were optimum. As evidenced from Figure 3A,B, with an increase in voltage from 8 to 10, there is an increase in *ΔE_(r-o)_* and a decrease in *ΔE_(p-o)_*, whereas with a further increase from 10 to 12, a decrease in *ΔE_(r-o)_* and a slight increase in *ΔE_(p-o)_* was observed. An increase in flow rate caused a slight increase in *ΔE_(r-o)_* but a decrease in *ΔE_(p-o)_*. With an increase in distance from 10 to 15 cm, a slight decrease in *ΔE_(r-o)_* was observed, but it rapidly decreased with an increase in distance to 15 cm. Horuz and Belibagli [52] reported the successful production of electrospun gelatin nanofibers using the smaller the better approach for fiber diameter using Taguchi’s methodology. They observed that the optimum values were 18 kV (voltage), 15 μL/min (flow rate), and 12.5 cm (distance) to obtain nanosized fibers. Similarly, using the Taguchi design, Albetran and others [57] optimized electrospun polyvinylpyrrolidone and titanium dioxide nanofibers and reported the optimum combination of flow rate, distance, and applied voltage as 1 mL/h, 11 cm, and 18 kV, respectively.

### 3.4. Morphology of Electrospun Tags

The microstructure of the kappa-carrageenan-based electrospun tag at scales of 10 µm and 100 µm is shown in Figure 4A. It reveals the nonwoven fiber structure of the electrospun tag. Fibers were observed to be continuous and arranged randomly. Average fiber diameter ranged between 1 and 2 μm. Smaller fiber size helps in providing a large surface area for electrospun tags, due to which a smaller amount of tag components could be used to fabricate electrospun tags. In our previous study, Deshwal and coworkers [7] reported that to prepare kappa-carrageenan-based leak indicator film of 1 m^2^ area, 1667 mL of casting solution is required. However, here we observed that only 143 mL is required to fabricate a similar area tag using electrospinning. Lu and coworkers [50] observed that smooth fibers of sodium alginate and PEO with a mean diameter of 228 nm could be fabricated with the two used in equal ratios. They reported that with doubling the amount of PEO fiber, diameter increased to 266 nm, which was attributed to the increased viscosity of the solution. Horuz and Belibagli [52] reported that gelatin-based electrospun nanofibers showed smooth and homogeneously distributed fiber diameters ranging between 76 and 288 nm.

### 3.5. Photo-Activation and Recovery Times of Electrospun Tag

The effects of exposure to oxygen on the physical appearance, photo-activation, and recovery times of the kappa-carrageenan-based electrospun tag are shown in Figure 4B. In the presence of ambient air (21% oxygen), the UV-activation (photo-activation) time of the tag was observed to be 7 min, and the recovery time was 7.67 h. The photo-activation and recovery times were found to be very high compared with kappa-carrageenan and methylene blue-based films, i.e., photo-activation (3–6 min) and recovery times (1 h) reported by Deshwal and coworkers [7]. Vu and Won [54] reported photo-activation and recovery times of iota-carrageenan-methylene blue-based indicator film mat as 4 min and 8 h, respectively. They attributed the high recovery time to the inherent hydrophilic property of carrageenan as hydrophobic films delay the formation of the charged methylene blue (MB^+^) and hydroxyl (OH^−^) species for color recovery.

The photo-activation and recovery time of the kappa-carrageenan-based electrospun tag at low oxygen concentration levels (0.4%) were 7 min and 24.67 ± 0.58 h, respectively. Recovery time of the kappa-carrageenan-based tag increased upon exposure to low oxygen concentration. The color recovery rate is linearly dependent on oxygen concentration (0 to 100%) [58]. Lawrie and coworkers [59] also reported that the recovery rate of a hydroxyl ethyl cellulose-based UV-activated colorimetric oxygen sensor containing methylene blue, titanium oxide, and glycerol is directly proportional to the headspace oxygen concentration in the package. Mills and Hazafy [60] reported that a hydroxyl ethylene cellulose, methylene blue, glycerol, and nanocrystalline SnO_2_-based colorimetric oxygen indicator had photo-activation and recovery times of 10 min and 10 min, respectively. The researchers observed that the photo-activation rate depends upon the irradiance of ultraviolet light and predicted that the half-life of the initial color of the activated film is directly proportional to the oxygen content of headspace gas composition.

### 3.6. Moisture Sorption Isotherms

The equilibrium moisture contents (EMCs) of the oxygen indicator tag were determined in the water activity (*a_w_*) range of 0.11–0.92 at 25 °C. EMCs of the tag ranged between 2.11 and 86.63 g.100 g^−1^ on a dry basis. It was observed that the EMC of the indicator tags slowly increased to 0.5 *a_w_* (2.11–11.61 g.100 g^−1^), after which it increased rapidly (17.52–86.63 g.100 g^−1^). The moisture sorption curve of both the leak indicator tags resembled the Type-III curve per BET classification. In a study on carrageenan-pectin-based biodegradable film, Alves and coworkers [61] reported a sharp increase in the EMC beyond 0.6. This increase was attributed to the hydrophilic nature of kappa-carrageenan similar to other biopolymers such as cellulose and gluten. Moisture sorption isotherm of most biopolymers is concave in shape [62], and in this study, as well, the carrageenan-based oxygen indicator tag’s sorption behavior was concave-shaped (please refer to supplementary information). The rapid rate of the moisture absorption might be credited to both the hydrophilic behavior of the polymer and the plasticizer’s high concentration [63]. Six models *viz.* BET, GAB, Ferro-Fontan, Peleg, Park, and D’Arcy, and Watt models were used to fit the moisture sorption behavior of the electrospun oxygen indicator tag. The derived parameters of the sorption models are presented in Table 2. It is evident from Table 2 that the Ferro-Fontan model best described the sorption data of the kappa-carrageenan-based electrospun oxygen indicator tag. The coefficient of the determination (*R*^2^) of the kappa-carrageenan-based tags was found to be 0.9897. The obtained *R*^2^ values were close to 1.0, which indicates ideal fit. Similarly, RMS% of the kappa-carrageenan-based tag was 7.14, which is less than 10, meeting the criteria for the best fit model. *p* was close to 5, i.e., 6.138, for the kappa-carrageenan-based tag. The pattern of residuals between actual values and the best fit Ferro-Fontan model is shown in Figure 5. Galus and Lenart [64] reported that sodium alginate/pectin-based antimicrobial films have a sigmoidal shape, as did Fan and coworkers [65] reporting on pectin/sodium alginate/xanthan gum-based films, and the Peleg model with a *p* value of 3.510 showed the best fit. Alves and coworkers [66] reported that the best-fit model for carrageenan/pectin film with mica flakes was GAB. Further, they reported that water entering the carrageenan/pectin-based composite film acts as a diffusion species and plasticizer, which results in loosening of the polymer matrix at a higher *a_w_.* Chen and coworkers [67] reported that the nonlinear nature of water absorption by sensor films was attributed to the absorption of water at hydrophilic sites, which brings structural changes to films.

### 3.7. Migration of Electrospun Indicator Tags

Global or overall migration into different food simulants revealed the maximum migration in 10% ethanol (81.33 ± 1.41 mg/dm^2^) and the minimum in *n*-heptane (7.10 ± 0.90 mg/dm^2^). Except for *n*-heptane, the migration exceeded the limits prescribed by the Bureau of Indian Standards [46]. This high amount of migration of leak indicator tag components into food simulants such as distilled water, 3% acetic acid, 10% ethanol, and 50% ethanol may be due to the hydrophilic nature of the components. Migration of the kappa-carrageenan-based electrospun tag was found to be similar to the kappa-carrageenan-methylene blue-based oxygen sensor film reported by Deshwal and coworkers [7]. Vu and Won [54] studied the leaching-resistant effect of a carrageenan-based colorimetric oxygen indicator and revealed that a negative charge on the carrageenan due to the sulfate group helps in preventing the migration. In the present study, however, the kappa-carrageenan-based tag dissolved, possibly due to the presence of a higher amount of PEO, which makes the tag more hydrophilic due to the presence of two side hydroxyl groups.

### 3.8. Storage Stability of Electrospun Tag

Visual appearances of stability of the kappa-carrageenan-based electrospun tag during storage as affected by lightness or darkness, temperature (5 °C and 25 °C), and relative humidity (65% and 95%) are presented in Figure 6A, and the total tag color differences (*ΔE*) are presented in Figure 6B. Figure 6A reveals that when tags were stored under light conditions, there was a change in color and visual appearance that could be clearly distinguished by the human eye after the 15th day, whereas there were no visual color changes at the end of 60 days when stored under dark conditions. Figure 6B reveals that the *∆E* of the kappa-carrageenan-based tag increased from 0 to 21.13 ± 0.58 on the 60th day when stored under light conditions and increased from 0 to 3.89 ± 0.85 only at the end of 60 days of storage under the dark condition. It has been reported in the literature that *ΔE* of more than 5.0 could be easily detected by human eyes and that *ΔE* of more than 12 indicates a distinguishable image [55]. This was evident in the present work. The kappa-carrageenan-based tag was found to be stable up to 60 and 5 days, respectively, when stored under dark and light conditions.

Further, Figure 6A shows that there is no change in visual color of the electrospun tag, which could be distinguished by the human eye at the end of 60 days at 5 °C and 25 °C under dark conditions. Figure 6B also reveals that when the tags were stored at 5 °C, *∆E* of the tag increased from 0 to 3.06 ± 0.51 at the end of the 60th day and from 0 to 3.89 ± 0.85 during the same period when stored at 25 °C under dark conditions. Both of these values are below the reference value (5.00), and the tags were found to be stable up to 60 days when stored at 5 °C and 25 °C. No visual changes were recorded in the tags stored under 65% relative humidity (RH), but at higher RH (95%), tags were found to shrink and lose structural integrity within 30 min (Figure 6A). This distortion in physical stability could be attributed to the hydrophilic nature of biopolymers and PEO used as one of the leak indicator components. The *∆E* of the tags stored at 65% RH increased from 0 to 4.59 ± 0.46 during the storage period (Figure 6B). Lu and coworkers [50] also reported quick dissolution of sodium alginate/PEO in water. Therefore, to store the developed electrospun kappa-carrageenan-based leak indicator tag, the best conditions were found to be under dark conditions at 65% RH either 5 °C or 25 °C. It can be concluded that under these conditions, the electrospun tags could be stored for up to 60 days.

## 4. Conclusions

The present study attempted to develop an electrospun kappa-carrageenan-based UV-activated smart oxygen leak indicator tag. The viscosity of the electrospinning solution was found to decrease with an increase in temperature (25, 35, and 45 °C). Power law was found to be the best-fit rheological model at all the temperatures. The electrospinning solutions showed shear thinning behavior at all temperatures, and the electrical conductivity of the solutions increased with an increase in temperature from 20 to 40 °C. The optimized parameters for the kappa-carrageenan-based leak indicator tag were voltage 8 kV, flow rate 0.4 mL/h, and distance 10 cm. The fabricated tags recorded a photo-activation time of 7 min and recovery time of 7.67 ± 0.24 h in presence of ambient air. The photo-activation and recovery time of the kappa-carrageenan-based electrospun tag at a low oxygen concentration (0.4%) was 7 min and 24.67 ± 0.58 h. The equilibrium moisture content of the tags at *a_w_* of 0.92 was 88.25 g/100 g on a dry matter basis. The Ferro-Fontan model was found to best describe the moisture sorption behavior of the electrospun tag. It was observed that the kappa-carrageenan-based tag was stable for 60 days at 25 °C and 65% RH when stored in dark conditions. However, further studies are required to reduce the overall or global migration of tag components into food simulants. It can be concluded that a kappa-carrageenan-based electrospun smart oxygen-indicating tag could find applications in detecting the presence of residual oxygen content in MAP foods, the seal integrity of MAP foods, the purity of nitrogen and carbon dioxide gas cylinders, etc., through visual colorimetric changes.

## Figures and Tables

**Figure 1 polymers-14-02108-f001:**
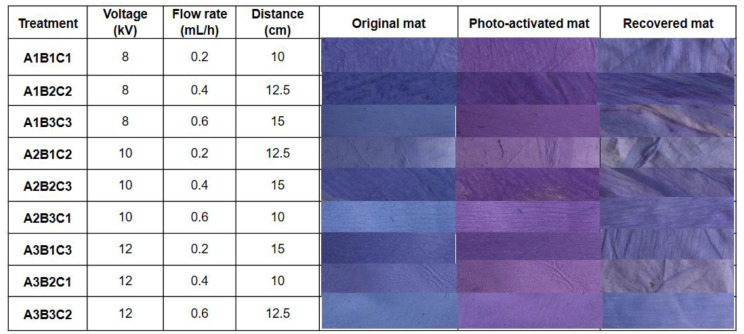
Experimental design for the selected factors with their levels and the visual appearance of the electrospun tags as affected by UV-light treatment and later exposure to ambient air.

**Figure 2 polymers-14-02108-f002:**
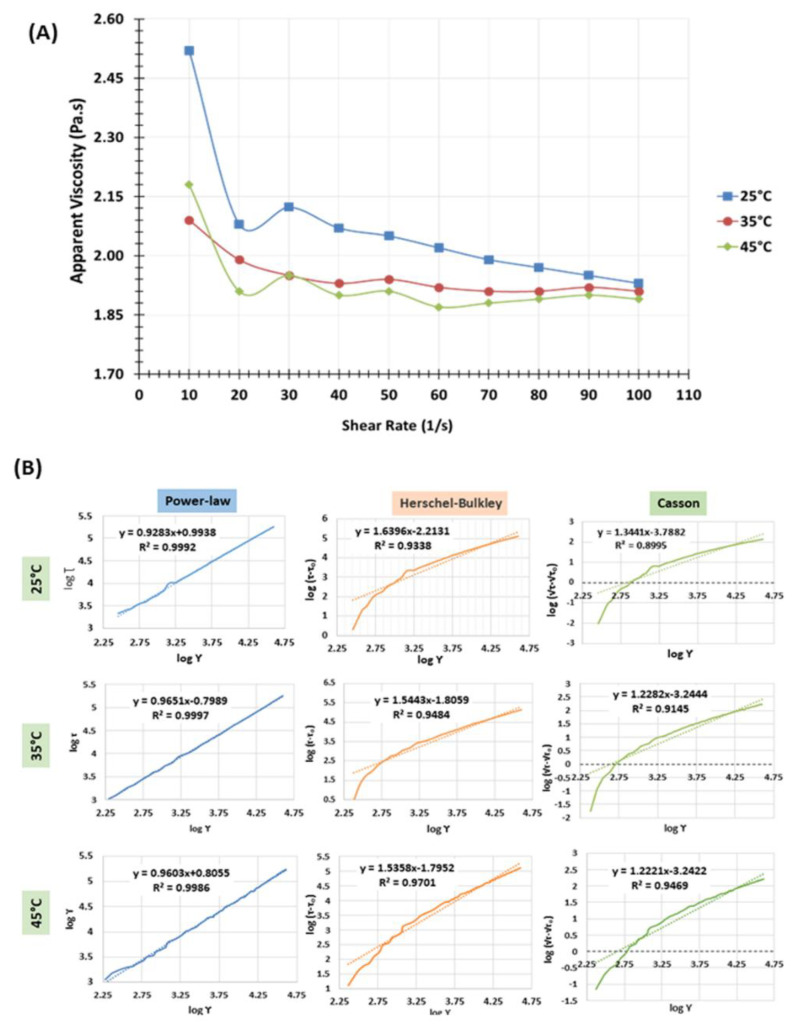
Rheological models of kappa-carrageenan-based electrospinning solution. (**A**): Apparent viscosity of electrospinning solution at three temperatures; (**B**): Fitting of mathematical models to describe the flow behavior of electrospinning solutions at three temperatures.

**Figure 3 polymers-14-02108-f003:**
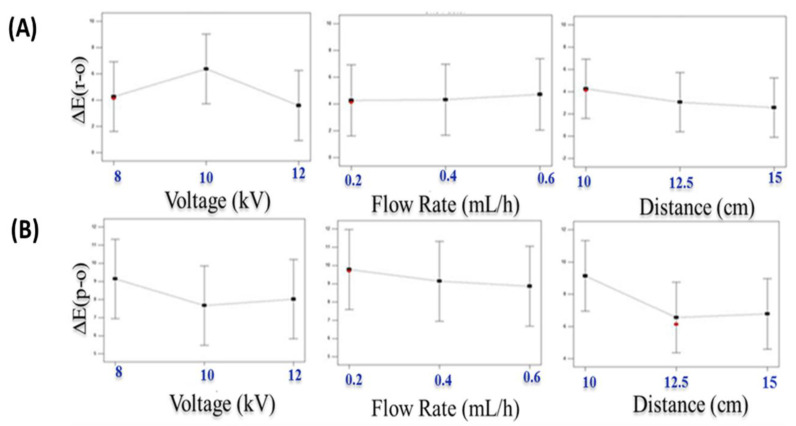
Effects of electrospinning process variables on total color difference (*∆E*) values of the electrospun tags. ((**A**): *∆E_(r-o)_*; (**B**): *∆E_(p-o)_*; subscripts: o-original; p-photo-activated; r-recovered).

**Figure 4 polymers-14-02108-f004:**
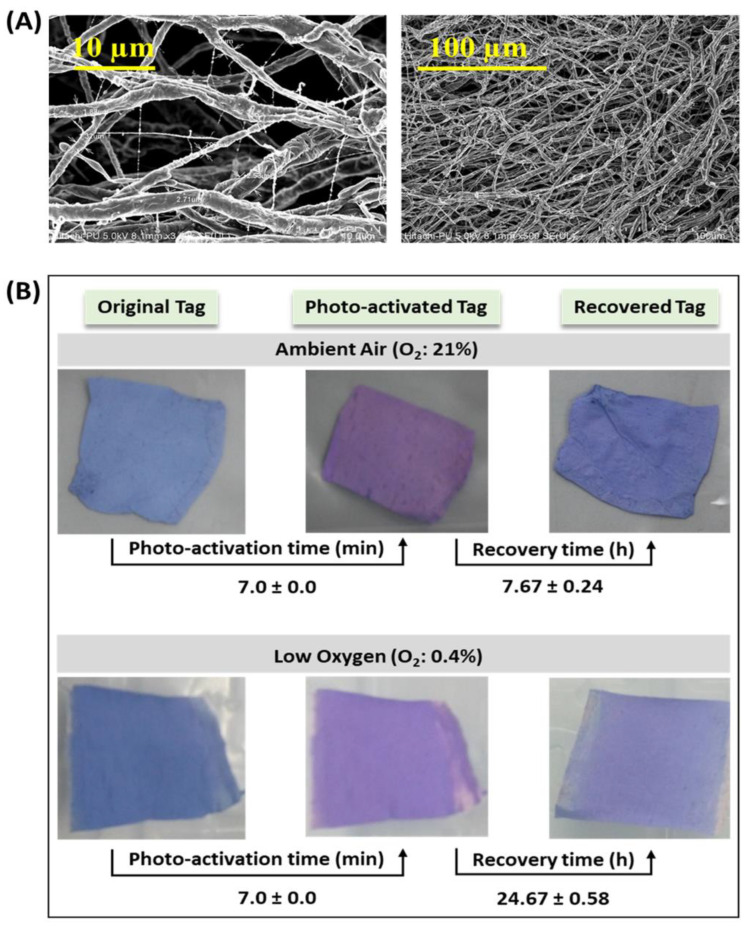
Microstructure and appearance of electrospun tag as affected by ultraviolet light activation: (**A**) microstructure; (**B**) original, photo-activated, and recovered tags.

**Figure 5 polymers-14-02108-f005:**
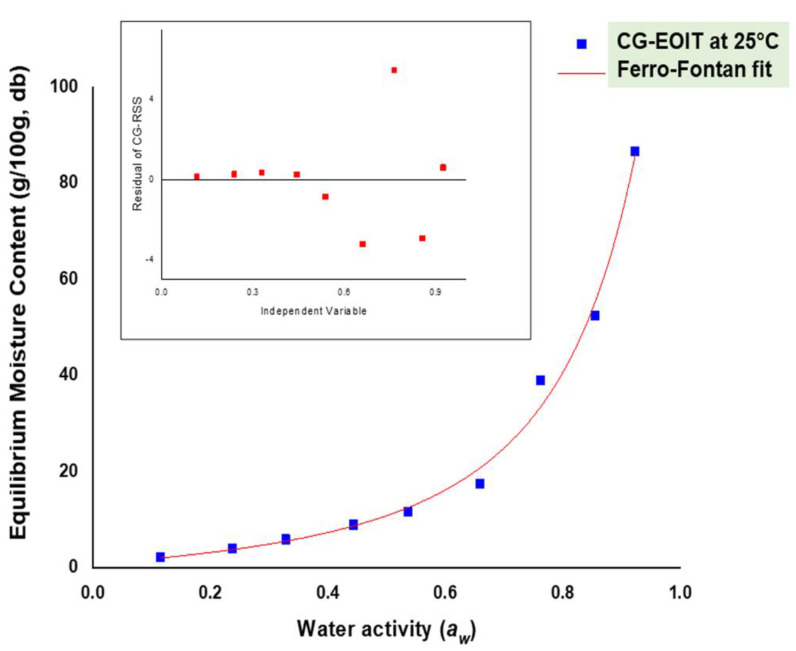
The moisture sorption isotherm of the kappa-carrageenan-based electrospun oxygen-indicating tag (CG-EOIT) and best-fit model (residuals pattern is presented in inset).

**Figure 6 polymers-14-02108-f006:**
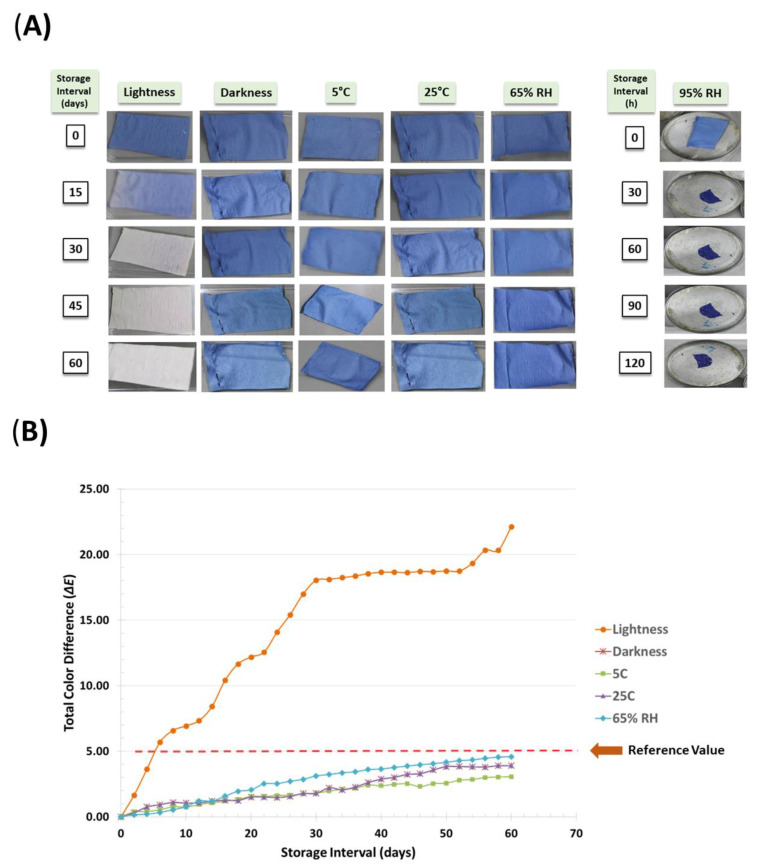
Storage stability of the kappa-carrageenan-based electrospun oxygen-indicating tag. (**A**) Images of the oxygen indicating tags stored under different conditions with their visual appearances; (**B**) The changes in total color difference values of oxygen indicating tags during storage.

**Table 1 polymers-14-02108-t001:** Effects of different electrospinning process conditions on the instrumental color values of the tag.

Treatment	Lightness			Redness			Blueness			*∆E_(r-o)_*	*∆E_(p-o)_*
	*L_o_**	*L_p_**	*L_r_**	*a_o_**	*a_p_**	*a_r_**	*b_o_**	*b_p_**	*b_r_**		
A1B1C1	16.00 ± 0.77 ^bA^	15.74 ± 1.05 ^cdA^	15.93 ± 0.79 ^bcA^	5.65 ± 1.14 ^dA^	10.68 ± 1.24 ^abB^	5.06 ± 1.38 ^bcA^	−16.23 ± 0.99 ^abA^	−15.94 ± 0.52 ^aA^	−14.26 ± 0.82 ^aB^	2.71 ± 0.60 ^abA^	5.08 ± 0.24 ^aB^
A1B2C2	12.13 ± 0.78 ^aAB^	11.64 ± 0.90 ^aA^	13.38 ± 0.41 ^aB^	4.44 ± 0.82 ^dA^	10.03 ± 0.97 ^abB^	5.26 ± 0.34 ^bcA^	−16.56 ± 1.26 ^abA^	−15.79 ± 1.08 ^aA^	−15.06 ± 0.28 ^aA^	2.53 ± 0.32 ^aA^	5.89 ± 0.79 ^abB^
A1B3C3	16.00 ± 0.73 ^bA^	14.85 ± 0.80 ^bcA^	16.44 ± 0.93 ^bcA^	4.71 ± 0.81 ^dA^	10.79 ± 0.49 ^abB^	5.21 ± 0.66 ^bcA^	−14.79 ± 1.09 ^bcA^	−14.74 ± 1.02 ^abA^	−10.35 ± 0.85 ^bB^	4.69 ± 0.79 ^cA^	6.24 ± 0.55 ^abB^
A2B1C2	16.54 ± 0.96 ^bA^	18.50 ± 0.88 ^eB^	18.82 ± 1.00 ^dB^	3.29 ± 0.40 ^bcA^	8.91 ± 0.91 ^aB^	4.15 ± 0.72 ^abcA^	−13.53 ± 0.99 ^cA^	−12.41 ± 1.68 ^cA^	−11.12 ± 0.72 ^bA^	3.76 ± 0.35 ^abcA^	6.14 ± 0.80 ^ab^
A2B2C3	13.38 ± 0.52 ^aA^	13.19 ± 0.95 ^abA^	14.68 ± 1.23 ^abA^	4.82 ± 0.97 ^dA^	11.73 ± 0.83 ^bB^	5.5 ± 1.31 ^cA^	−15.73 ± 0.97 ^abAB^	−16.29 ± 0.84 ^aA^	−14.32 ± 0.95 ^aB^	2.54 ± 0.73 ^aA^	7.02 ± 0.28 ^bB^
A2B3C1	13.77 ± 0.88 ^aA^	14.78 ± 0.94 ^bcA^	16.57 ± 0.62 ^cB^	5.32 ± 0.72 ^dA^	10.06 ± 2.07 ^abB^	5.18 ± 0.84 ^bcA^	−17.12 ± 1.21 ^aA^	−15.53 ± 1.10 ^aAB^	−13.5 ± 0.67 ^aA^	4.62 ± 0.46 ^cA^	5.23 ± 0.99 ^aA^
A3B1C3	17.08 ± 0.94 ^bA^	18.38 ± 1.31 ^eAB^	20.02 ± 1.57 ^dB^	5.24 ± 0.78 ^dA^	11.09 ± 1.64 ^abB^	3.82 ± 0.36 ^abA^	−16.5 ± 0.76 ^abA^	−13.35 ± 1.21 ^bcB^	−10.09 ± 1.74 ^bC^	7.47 ± 1.42 ^dA^	6.97 ± 1.00 ^bA^
A3B2C1	18.90 ± 1.71 ^cA^	17.33 ± 1.15 ^deA^	19.09 ± 1.38 ^dA^	1.85 ± 0.34 ^aA^	10.18 ± 0.92 ^abB^	3.21 ± 0.85 ^aA^	−15.77 ± 0.83 ^abAB^	−16.53 ± 0.70 ^aA^	−14.29 ± 1.42 ^aB^	3.17 ± 1.10 ^abA^	8.53 ± 0.61 ^cB^
A3B3C2	19.78 ± 0.91 ^cA^	18.82 ± 1.10 ^eA^	18.35 ± 0.36 ^dA^	2.47 ± 0.6 ^abA^	11.91 ± 1.20 ^bB^	5.56 ± 0.67 ^cA^	−17.03 ± 1.38 ^aA^	−15.91 ± 0.82 ^aA^	−15.09 ± 0.55 ^aA^	4.14 ± 0.44 ^bcA^	9.70 ± 0.53 ^cB^

*L_o_**: Lightness value of original mat; *L_p_**: Lightness value of photoactivated mat; *L_r_**: Lightness value of recovered mat, *a_o_**: Redness to greenness value of original mat; *a_p_**: Redness to greenness value of photoactivated mat; *a_r_**: Redness to greenness value of recovered mat; *b_o_**: Yellowness to blueness value of original mat; *b_p_**: Yellowness to blueness value of photoactivated mat; *b_r_**: Yellowness to blueness value or recovered mat; *∆E_(r-o)_*: Total color difference value between recovered and original mats; *∆E_(p-o)_*: Total color difference value between photoactivated and original mats. ^ABC^ Meanvalues within a color attribute (row) with at least one similar superscript do not differ significantly (*p* > 0.05); ^abcde^ Mean values within a column with at least one similar superscript do not differ significantly (*p* > 0.05).

**Table 2 polymers-14-02108-t002:** Moisture sorption models of electrospun tags and their calculated values.

Model	Parameter	Estimated Values	Model	Parameter	Estimated Values
**BET** **(*a_w_* = 0.1–0.5)**	*M* _o_	7.37	Peleg (*a_w_* = 0.1–0.9)	*A*	112.39
	*C* _b_	2.43		*B*	23.09
	*R* ^2^	0.9979		*C* _1_	6.96
	*P*	**0.489**		*C* _2_	1.24
	*RMS%*	**2.46**		*R* ^2^	0.9859
				*P*	4.82
				*RMS%*	11.20
GAB (*a_w_* = 0.1–0.9)	*K*	0.74	D’Arcy & Watt (*a_w_* = 0.1–0.9)	*K* _1_	2.81 × 10^−6^
	*C*	2.13		*K* _2_	−228.30
	*M* _o_	9.84		*K* _3_	−5.19
	*R* ^2^	0.9896		*K* _4_	0.92
	*χ* ^2^	8.51		*K* _5_	15.88
	*P*	6.82		*R* ^2^	0.9844
	*RMS%*	16.60		*P*	7.07
				*RMS%*	17.75
**Ferro-Fontan** **(*a_w_* = 0.1–0.9)**	*A*	1.20	**Park** **(*a_w_* = 0.1–0.9)**	*A*	−0.489
	*B*	3.49		*B*	−7.19
	*C*	0.58		*H*	19.69
	*R* ^2^	**0.9897**		*K*	17.31
	*P*	**6.138**		*n*	6.61
	*RMS%*	**9.16**		*R* ^2^	**0.9823**
				*P*	**4.94**
				*RMS%*	**9.09**

*R^2^* = Coefficient of determination, *P* = Percent mean deviation error, *RMS%* = Root mean square error percent (values highlighted in bold are the best-fit models).

## Data Availability

Not applicable.

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
