# Peer review of "Electrospun Smart Oxygen Indicating Tag for Modified Atmosphere Packaging Applications: Fabrication, Characterization and Storage Stability"

_polymers, 2022, doi:10.3390/polym14102108_

Round 1

Reviewer 1 Report

The present study attempted to develop an electrospun kappa-carrageenan-based UV-activated smart oxygen leak indicator tag. Power law was found to be the best fit rheological model at all the temperatures. Electrospinning solutions showed shear thinning behavior at all temperatures. The optimized parameters for the kappa-carrageenan-based leak indicator tag were successfully achieved. The kappa-carrageenan-based electrospun smart oxygen indicating tag could find applications in detecting the presence of residual oxygen content in MAP foods, seal integrity of MAP foods, purity of nitrogen and carbon dioxide gas cylinders, etc. through visual colorimetric changes. The study is comprehensive, which can be published in Polymers.

Author Response

Thank you for your positive evaluation.

The article's grammar and punctuation have been corrected as far as possible. Details may be found in the manuscript as the changes have been made in track changes mode.

Reviewer 2 Report

Summary 

The manuscript entitled “Electrospun Smart Oxygen Indicating Tag for Modified Atmosphere Packaging Applications Fabrication, Characterization and Storage Stability’’ by Kataoka et. al shows the development and characterization of electrospun ultraviolet light-activated colorimetric oxygen indicator tag for MAP foods.

General comments

In general, the work is quite accurate and well presented as  Polymers articles should be anyway the presented results are certainly of interest to readers of this journal. The article style is not fully correct, and it should be reviewed on a few points. Thus, I believe that the text needs some technical adjustments to be published. Therefore, I recommend that this manuscript can be published in Polymers after major revision.

Specific comments

Going into details on the specific issues, here some comments are reported:

- the article's grammar, punctuation, and style are quite poor, and the manuscript needs to be proofread.

- The authors took for granted that everybody knows the basics of nanomaterial-based sensing, but this is not true as the matter of fact. A full paragraph should be dedicated to it in the Introduction, including the main concept and application of polymer-based nanostructured materials in the sensing fields, including pathogens [https://doi.org/10.1016/j.coelec.2021.100848] and molecule biosensing [https://doi.org/10.1038/s41427-022-00365-9]. Please mention these brand-new impactful articles in the Introduction section.

- the manuscript should follow a logical path. It is not clear why the electrospun mat characterization section is put before the fabrication in Materials and Methods

- Figure 4A. The scale bars are not visible

- some basic characterization of materials (e.g., FT-IR, XRD, and DSC) have not been performed.

-UV irradiation can strongly affect the stability of the electrospun materials. SEM and GPC analyses of the as-spun mats, as well as the same fibrous materials after different irradiation cycles, are necessary.

Conclusion

The topic of this manuscript falls within the scope of Polymers. I like the concept and material development/characterization proposed in this paper. I believe the article is of sufficient novelty to meet the Polymers publication standard after major revision.

Author Response

Reviewer-2 Comments

Action taken

1.      The article's grammar, punctuation, and style are quite poor, and the manuscript needs to be proofread.

2.      The authors took for granted that everybody knows the basics of nanomaterial-based sensing, but this is not true as the matter of fact. A full paragraph should be dedicated to it in the Introduction, including the main concept and application of polymer-based nanostructured materials in the sensing fi elds, including pathogens and molecule biosensing. Please mention these brand-new impactful articles in the Introduction section.

3.      The manuscript should follow a logical path. It is not clear why the electrospun mat characterization section is put before the fabrication in Materials and Methods.

4.      Figure 4A. The scale bars are not visible

5.      Some basic characterization of materials (e.g., FT-IR, XRD, and DSC) have not been performed.

6.      UV irradiation can strongly affect the stability of the electrospun materials.

1.      The article's grammar and punctuation have been corrected as far as possible. Details may be found in the manuscript as the changes have been made in track changes mode.

2.      A separate paragraph on nanomaterials-based sensing has been included in the introduction section between 97 and 104 lines. Four new references have been cited after 12th reference and accordingly, other references numbers have been  updated. However, references suggested by Reviewer-2 couldn’t be included as they were not directly relevant to the current topic.

3.      As far as possible, the work was presented in a logical manner in the manuscript. In the Materials and Methods section (2.0), Section 2.1 describes the characterization of electrospinning solution, section 2.2 describes the optimization of the process parameters of the electrospinning technique, section 2.3 characterization of electrospun tag, and section 2.4 describes the storage stability aspects.

4.      Figure 4A is provided separately to show the scale bars. It may be included as supplementary information.

5.      Most appropriate aspects of biopolymer-based electrospun materials characterization have been carried out.

6.      In the present study, the non-UV treated electrospun mats were subjected to storage stability and the intention was to check the stability of the electrospun mats before application. UV irradiation may affect the stability of the electrospun materials after application. But this needs to be studied separately.

Round 2

Reviewer 2 Report

I should point out that I reviewed a few hundred scientific articles, and I have never experienced a situation like this.

The Authors did not understand that the comments of Reviewers are directed toward helping them reach the best possible results. Panawar et al. did their best to avoid any revision of the manuscript and frankly speaking, I view sentences like ‘’’ All the comments we received on this study have been taken into account in improving the quality of the article, and we present our reply to each of them separately. We believe that the Editor and reviewers’ suggestions have been very helpful in improving the manuscript. The manuscript are enclosed and all the changes have been underlined.’’’ written by the Authors as a sort of mockery.

Going into the details of my comment:

  1. The article was not proofread. The authors only added some articles from time to time, while a severe revision is needed. The flow of the text is in an awful state, and it is, from time to time, unreadable. I would say that my recommendation has been followed for no more than 20%
  2. I asked to add a paragraph about sensing, in general, to open the story to readers who are not experts. The authors add a short text, but it is not very connected to the suggestion. I would say that my recommendation has been followed for no more than 50%
  3. The logic of the reply could have a sense, but my comment should trigger the author's thinking. Is ’’Characterization of electrospinning solution’’ the right title for this subsection? I do not think so. It looks like simply part of the optimization and/or preliminary study. I would say that my recommendation has been followed for no more than 50%
  4. The authors added the scale bares as I requested, but they did not remove the machine-generated piece of information as well as all the tags used to measure fiber diameters present in Fig 4A. I would say that my recommendation has been followed for no more than 50%
  5. The authors did not take into consideration all these comments, which is very important. I would say that my recommendation has been followed for no more than 0%
  6. The authors did not take into consideration all these comments, which is absolutely needed. I would say that my recommendation has been followed for no more than 0%

In conclusion, the revision should be done from scratch again. The article is far to be publishable in its present form. If Panwar et al. are not willing to revise the manuscript, I would suggest rejecting it.